# Dynamic allocation of limited memory resources in reinforcement learning

**Nisheet Patel**[*]
Department of Basic Neurosciences
University of Geneva
nisheet.patel@unige.ch

**Luigi Acerbi**
Department of Computer Science
University of Helsinki
luigi.acerbi@helsinki.fi

**Alexandre Pouget**
Department of Basic Neurosciences
University of Geneva
alexandre.pouget@unige.ch

## Abstract

Biological brains are inherently limited in their capacity to process and store information, but are nevertheless capable of solving complex tasks with apparent ease. Intelligent behavior is related to these limitations, since resource constraints drive the need to generalize and assign importance differentially to features in the environment or memories of past experiences. Recently, there have been parallel efforts in reinforcement learning and neuroscience to understand strategies adopted by artificial and biological agents to circumvent limitations in information storage. However, the two threads have been largely separate. In this article, we propose a dynamical framework to maximize expected reward under constraints of limited resources, which we implement with a cost function that penalizes precise representations of action-values in memory, each of which may vary in its precision. We derive from first principles an algorithm, Dynamic Resource Allocator (DRA), which we apply to two standard tasks in reinforcement learning and a model-based planning task, and find that it allocates more resources to items in memory that have a higher impact on cumulative rewards. Moreover, DRA learns faster when starting with a higher resource budget than what it eventually allocates for performing well on tasks, which may explain why frontal cortical areas in biological brains appear more engaged in early stages of learning before settling to lower asymptotic levels of activity. Our work provides a normative solution to the problem of learning how to allocate costly resources to a collection of uncertain memories in a manner that is capable of adapting to changes in the environment.

## 1 Introduction

Reinforcement learning (RL) is a powerful form of learning wherein agents interact with their environment by taking actions available to them and observing the outcome of their choices in order to maximize a scalar reward signal. Most RL algorithms use a value function in order to find good policies [1, 2] because knowing the optimal value function is sufficient to find an optimal policy [3, 4]. However, RL models typically assume that agents can access and update the value function for each state or state-action pair with nearly infinite precision. In neural circuits, however, this assumption

---

[*]Current address: Département des neurosciences fondamentales, Université de Genève, CMU, 1 rue Michel-Servet, 1206 Genève, Switzerland. Alternative e-mail: nisheet.pat@gmail.com.

is necessarily violated. The values must be stored in memories which are limited in their precision, especially in smaller brains in which the number of neurons can be severely limited [5].

In this work, we make such constraints explicit and consider the question of what rationality is when computational resources are limited [6, 7]. Specifically, we examine how agents might represent values with limited memory, how they may utilize imprecise memories in order to compute good policies, and whether and how they should prioritize some memories over others by devoting more resources to encode them with higher precision. We are interested in drawing useful abstractions from small-scale models that can be applied generally and in investigating whether brains employ similar mechanisms.

We pursue this idea by considering memories that are imprecise in their representation of values, which are stored as distributions that the agent may only sample from. We construct a novel family of cost functions that can adjudicate between maximizing reward and limited coding resources (Section 2). Crucially, for this general class of resource-limited objective functions, we derive how an RL agent can solve the resource allocation problem by computing stochastic gradients with respect to the coding resources. Further, we combine resource allocation with learning, enabling agents to assign importance to memories dynamically with changes in the environment. We call our proposed algorithm the Dynamic Resource Allocator (DRA), which we apply to standard tasks in RL in Section 3 and a model-based planning task in Section 4.[1]

## 1.1 Related work

Our work is related to previous research within the paradigm of bounded rationality [6, 7] on two accounts. First, previous studies have considered capacity-limited agents that trade reward to gain information [8–11], but did so in a way that abstracts away the underlying costs, and therefore cannot disambiguate the effects of limited storage capacity from other energetic or computational limitations. Second, a separate line of work makes the limitations in memory explicit [12–14] like we do, but such studies restrict their analyses to simple working memory tasks, such as reproducing observed stimuli or delayed recall, and lack a general-purpose, dynamical framework for decision-making. Our work is also ideologically related to that of others looking into prioritized replay of memories, both in RL [15] and neuroscience [16], with the important difference that these studies do not make the uncertainty in agents' memories explicit as we do. Moreover, agents in Mattar and Daw [16] replay memories to update their q-value estimates and hence their policy. Thus, prioritization of memories in [16] stems only due to incomplete learning and their results would not hold for well-trained agents, in contrast to our work where these effects are driven by a limited capacity constraint.

Other groups have proposed alternative approaches to deal with memory limitations in RL, such as using regularization (SAC [17]), or using neural networks for representing policy and value functions, and even compressing state representations with graph-Laplacian [18]. Our work is meant to complement these previous studies. SAC, for instance, directly penalizes the policy entropy while maximizing reward to encourage exploration. In DRA, we penalize precise representations of q-values instead. The use of a compressed graph-Laplacian [18], on the other hand, hints at yet another problem involving efficient use of memory – compact representation of states (e.g., chunking) – which we plan to combine with our approach in future work. On the technical side, DRA is related to O'Donoghue et al.'s uncertainty-based exploration in the manner of decoupling updates to different moments of the value distribution [19]. Finally, it is worth noting that our work fundamentally differs from previous work in RL applied to *external* resource allocation [20, 21] in that we are using RL to optimize the computational resources *of the agent itself*.

## 2 Background and details

### 2.1 Environment and agent's memories

We consider problems that can be described by a Markov Decision Process (MDP) characterized by a quadruple $\langle \mathcal{S}, \mathcal{A}, \mathcal{R}_a, \mathcal{P}_a \rangle$, where $\mathcal{S}$ is a finite set of states that describe the environment, $\mathcal{A}$ is a finite set of actions available to the agent, $\mathcal{R}_a(s, s')$ is the immediate reward received after taking action $a$

in state $s$ and transitioning to state $s'$, and $\mathcal{P}_a = p(s'|s,a)$ denotes the dynamics of the environment, *i.e.* probability of transitioning from state $s$ to $s'$ after taking action $a$ [22].

For simplicity, we assume that agents have a model of the world, though this assumption is not necessary except in model-based planning tasks such as the one in Section 4. We also assume that the agents perfectly observe the states. However, agents may only represent the value of each state-action pair, the *q-value*, with finite precision. This means that the q-values are represented as distributions rather than point estimates, and agents can only access samples from the distribution stored in memory but cannot access its parameters such as the mean and precision directly. We would like to emphasize that, for such agents, storing items in memory more precisely requires more resources.

Concretely, we define the agent's memory in a tabular form comprising the states, actions, immediate (mean) rewards, next states, and the corresponding imprecise q-value, represented here by a normal distribution with mean $\bar{q}_{sa}$ and variance $\sigma_{sa}^2$. Thus, each memory can be written as a quintuple $\langle s, a, r, s', \mathcal{N}(\bar{q}_{sa}, \sigma_{sa}^2) \rangle$, and the total number of memories equals $|\mathcal{S} \times \mathcal{A}|$. In this work, we assume the overall q-value distribution to be a multivariate normal with diagonal covariance matrix (that is, independent memories). However, our approach for allocating resources across a collection of uncertain memories can be generalized to arbitrary distributions.

## 2.2 Objective and policy

The goal of the agent is to maximize their expected sum of future rewards in a given task episode, subject to a cost of representing the q-values precisely. Biological agents pay such costs for recruiting more neurons for representation and computation of task-relevant statistics, or for creating the relevant synaptic connections between existing neurons for learning [23]. In general, our method can be applied to arbitrarily defined cost functions, but here we focus on a cost derived from neural and information-theoretical principles.

Biological agents must use a pre-existing population of neurons from a region of their brain where q-values are represented [24–26] to store their base q-value distribution in its connections and activity. We will refer to this base distribution as $P := \mathcal{N}(\bar{q}, \sigma_{\text{base}}^2 I)$. As brains learn to perform well on the task, they update the synaptic connections and may recruit more neurons if necessary to represent the q-value distribution in memory as $Q := \mathcal{N}(\bar{q}, \boldsymbol{\sigma}^2 I)$, with higher precision. In this work, we reason that the agent (and the brain) pays a cost proportional to the KL-divergence $D_{\text{KL}}(Q \parallel P)$, representing the information-theoretical cost for encoding deviations from the base distribution (*e.g.*, due to modified connectivity and neuronal activity).

Thus, the full objective that the agent is trying to maximize in each episode is:

$$\mathcal{F} := \mathbb{E}_\pi \left[ \sum_{t=0}^{\infty} \gamma^t r_{t+1} \middle| Q = \mathcal{N}(\bar{q}, \boldsymbol{\sigma}^2 I) \right] - \lambda D_{\text{KL}} \left( Q = \mathcal{N}(\bar{q}, \boldsymbol{\sigma}^2 I) \parallel P = \mathcal{N}(\bar{q}, \sigma_{\text{base}}^2 I) \right) \quad (1)$$

where $\gamma \in [0, 1]$ is the discount factor which we set to 1 in this article, the first term represents the expected reward for the MDP given the agent's memory distribution $Q$ and policy $\pi$, and the second term represents the cost function described above with a cost per *nat* equal to $\lambda \geq 0$.

Given that their memories are noisy, agents can only draw samples from their memory distribution of q-values. If the agent is greedy, they will then choose the action corresponding to the largest sampled q-value, effectively yielding the policy $\pi$:

$$\pi(a|s) = \Pr(a = \arg\max_{a'} \tilde{q}(s, a')) \qquad \text{with} \quad \tilde{q}(s, a') \sim \mathcal{N}\left(\bar{q}(s, a'), \sigma^2(s, a')\right). \quad (2)$$

This behavioral policy is also known as Thompson sampling [27, 28], which is often chosen deliberately for efficient exploration, but here, it is a consequence of having imprecise memories. In principle, agents could draw more than one sample from memory to increase the precision of their estimates of q-values, but we assume that drawing multiple *independent* samples would require additional time [29]. In this paper, we restrict our analyses to single Thompson samples at decision time, leaving a detailed analysis of the speed-accuracy trade-off [30] for future work.

Our key contribution in this work is providing the agent with control over the precision of each of its memories, the resource allocation vector $\boldsymbol{\sigma}$. Thus, the agent may allocate more resources to some memories than others so as to maximize the objective $\mathcal{F}$ defined in Eq. 1.

## 2.3 Maximizing the objective

First, we consider the scenario where the agent wishes to allocate resources optimally – by maximizing the objective $\mathcal{F}$ in Eq. 1 – when the mean q-values, $\bar{q}$, are fixed and known. To do so, we analytically derive a stochastic gradient for $\mathcal{F}$.

To compute the gradient of the first term of $\mathcal{F}$, *i.e.* the expected reward, we follow the approach as in the policy gradient theorem [31][22, section 13.2], with the difference being that our gradient is with respect to the resource allocation vector, $\boldsymbol{\sigma}$, instead of, for instance, the parameters of a policy network. In general, this gradient may be written as:

$$\nabla_\sigma \mathbb{E}_\pi \left[ \sum_{t=0}^\infty \gamma^t r_{t+1} \Big| Q = \mathcal{N}(\bar{\boldsymbol{q}}, \boldsymbol{\sigma}^2 I) \right] = \mathbb{E}_\pi \left[ \sum_{t=0}^\infty \Psi_t \nabla_\sigma \log \pi(a_t|s_t) \Big| Q = \mathcal{N}(\bar{\boldsymbol{q}}, \boldsymbol{\sigma}^2 I) \right] \quad (3)$$

where $\Psi_t$ can take many forms with different computational properties [32], including but not limited to:

$$\Psi_t = \begin{cases} \sum_{t'=t}^\infty \gamma^{t-t'} r_{t'+1} & \text{R-gradient (REINFORCE)} \\ \bar{q}(s_t, a_t) & \text{Q-gradient (mean q-value)} \\ A(s_t, a_t) & \text{A-gradient (\textit{advantage function}).} \end{cases} \quad (4)$$

In our case, we define the *advantage function* as $A(s_t, a_t) = \bar{q}(s_t, a_t) - |\mathcal{A}|^{-1} \sum_a \bar{q}(s_t, a)$ (see justification for using the mean $\bar{q}$ in Appendix A.3). In more complicated tasks that involve planning (*e.g.*, Section 4), agents may sample *multiple* future trajectories using the policy $\pi$, followed by performing a non-linear operation such as picking the maximal reward of all sampled trajectories. Thus, it would be inappropriate to use the advantage function, which only characterizes the expected reward until termination for a *single* trajectory (minus a baseline). In such scenarios, we replace $\Psi_t$ with the reward that results from planning, $\max_i(\sum_{t=0}^T r_{t+1,i})$, where $i$ indexes planned trajectories and $T$ is the planning horizon, following the reasoning as in the original REINFORCE algorithm [31].

Eq. 3 allows us to compute an unbiased stochastic estimate of the gradient – the expectation on the right-hand side – via Monte Carlo, *i.e.* by sampling one trajectory or averaging over multiple sampled trajectories. Note that if the agents do not have a model of the world, they may simply store previously experienced sequences of states in a buffer (episodic memory) and use these stored trajectories instead of generating new ones with their model. Unless otherwise mentioned, we use $N_{\text{traj}} = 10$ trajectories to compute the stochastic approximation for the expectation.

We obtain an analytical approximation for $\nabla_\sigma \log \pi(a|s)$ (Eq. 3) for our policy by first reparametrizing our q-value distribution as Kingma and Welling prescribe for normal distributions [33], and then using soft Thompson sampling [27, 28], *i.e.* using *softmax* or *soft*-arg max instead of arg max in Eq. 2 (see Appendix A.1 for full derivation). This modification yields:

$$\frac{\partial}{\partial \sigma(s', a')} \log \pi(a|s) = \begin{cases} \beta\zeta_{sa}(1 - \pi(a|s)) & \text{for } s' = s, a' = a \\ -\beta\zeta_{sa'}\pi(a'|s) & \text{for } s' = s, a' \neq a \\ 0 & \text{for } s' \neq s \end{cases} \quad (5)$$

where $\zeta_{sa} \stackrel{\text{i.i.d.}}{\sim} \mathcal{N}(0, 1) \ \forall s \in \mathcal{S}, a \in \mathcal{A}$, and $\beta$ is the inverse-temperature parameter for *softmax*. Plugging this into Eq. 3 for each step in the sampled trajectories yields the stochastic gradient of the first term of $\mathcal{F}$, the expected reward, with respect to $\boldsymbol{\sigma}$.

Computing the gradient of the second term of $\mathcal{F}$, the cost, is straightforward (see Appendix A.2):

$$\frac{\partial}{\partial \sigma_{sa}} D_{\text{KL}}(Q = \mathcal{N}(\bar{\boldsymbol{q}}, \boldsymbol{\sigma}^2 I) \parallel P = \mathcal{N}(\bar{\boldsymbol{q}}, \sigma_{\text{base}}^2 I)) = \frac{\sigma_{sa}}{\sigma_{\text{base}}^2} - \frac{1}{\sigma_{sa}}. \quad (6)$$

Now, the agent may iteratively update its resource allocation vector, $\boldsymbol{\sigma}$, as:

$$\boldsymbol{\sigma} \leftarrow \boldsymbol{\sigma} + \alpha\nabla_\sigma \mathcal{F} \quad (7)$$

where $\alpha > 0$ is a learning rate and $\mathcal{F}$ is the objective in Eq. 1 that depends on the resource allocation vector $\boldsymbol{\sigma}$, the stochastic gradient for which is given by Eqs. 3, 5, and 6.

Alternatively, we can maximize $\mathcal{F}$ via a black-box optimizer such as Covariance Matrix Adaptation Evolution Strategy (CMA-ES) [34, 35], which works well for up to hundreds of dimensions (memories, in our case). Results from CMA-ES provide us with a baseline for our gradient-based optimization in small-scale environments (Section 4), since both methods are able to find the optimal resource allocation at stationary state, *i.e.* when the mean q-values, $\bar{q}$, are fixed and known.

## 2.4 Dynamic allocation of limited memory resources

Real-world environments are rarely stationary. Thus, any agent with limited memory resources must assign importance dynamically to items in memory in a time-efficient manner. Our framework makes it is possible to do so by decoupling the updates for the mean and the variance of the memory distribution $Q$ (as, for instance, done by O'Donoghue et al. [19]). Agents can update the mean of $Q$ on the fly with any on-policy learning algorithm (*e.g.* SARSA [4] or expected SARSA [36]) and simultaneously update the variance as in Eq. 7. Combining these elements, we propose Algorithm 1, Dynamic Resource Allocator (DRA), to enable memory-limited agents to find good policies.

---

**Algorithm 1:** Dynamic Resource Allocator (DRA)

Set hyper-parameters $\boldsymbol{\theta} = (\alpha_1, \alpha_2, \beta, \gamma, \lambda)$
Initialize $\bar{\boldsymbol{q}}$, $\boldsymbol{\sigma}$, table of memories = $\langle s, a, r, s', \mathcal{N}(\bar{q}_{sa}, \sigma_{sa}^2 I) \rangle^{|\mathcal{S} \times \mathcal{A}|}$
**for** *episode* $k = 1, ..., K$ **do**
    $s \leftarrow s_0$
    **while** *s is not Terminal* **do**
        $a \leftarrow \pi(s, Q, \beta)$
        $s', r \leftarrow \text{Environment}(s, a)$
        $\delta \leftarrow r + \gamma \sum_{a'} \pi(a'|s') q(s', a') - q(s, a)$
        $q(s, a) \leftarrow q(s, a) + \alpha_1 \delta$
        $s \leftarrow s'$
    **end**
    Sample $N$ trajectories to compute:
$$\nabla_\sigma \mathbb{E}_\pi[\textstyle\sum_{t=0}^{\infty} \gamma^t r_{t+1}|Q] = \frac{1}{N} \sum_{n=1}^{N} [\sum_{t=0}^{\infty} \Psi_t \nabla_\sigma \log \pi(a_t|s_t)]$$
            where              $\Psi_t \leftarrow$ Eq. 4
                           $\nabla_\sigma \log \pi(a_t|s_t) \leftarrow$ Eq. 5
$$\boldsymbol{\sigma} \leftarrow \boldsymbol{\sigma} + \alpha_2 \left( \nabla_\sigma \mathbb{E}_\pi \left[ \textstyle\sum_{t=0}^{\infty} \gamma^t r_{t+1}|Q \right] + \lambda \nabla_\sigma D_{\text{KL}}(Q \parallel P) \right)$$
                       where $\nabla_\sigma D_{\text{KL}}(Q \parallel P) \leftarrow$ Eq. 6
**end**

---

# 3 Results on standard RL environments

## 3.1 2D Grid-world

First, we consider the grid-world adapted from Mattar and Daw [16] and depicted in Fig. 1a. The goal of the agent is to find the shortest route from the starting location indicated by the position of the rat to the cheese, since all transitions yield a reward -1 except reaching the cheese, which rewards 10 points. In each state, the agent can only choose to go *up*, *down*, *left*, or *right*, and if their intended action is blocked by an obstacle, that action leaves their position unchanged.

Our results show that the initial amount of resources agents can afford at the beginning of training has a critical influence on learning speed. We choose four different values of $\sigma_0$ (see Fig. 1b legend) to initialize $\sigma(s, a) = \sigma_0 \ \forall (s, a)$ for the memory distribution, and for each of them, we perform five optimization runs for 5000 episodes each. As shown in Fig. 1b, all $\sigma_0$ conditions eventually converge to the same solution (asymptotic $\sigma_*$ is largely independent of the initial $\sigma_0$), but starting with more resources (low $\sigma_0$) leads to faster learning, at a higher initial cost. In real-world animal experiments, this is a very common observation: more neurons are responsive to task-relevant variables in the early stages of training than when the animal has been well-trained [37, 38], suggesting that biological brains may indeed deliberately assign more resources early to enable quicker learning.

Further, we test the ability of DRA to dynamically reallocate resources by changing the rewards and transition structure of the task: we remove the obstacle directly adjacent to the cheese and re-position the cheese two states to the left after 3000 episodes in a separate experiment. This modification changes the shortest path to the cheese, and we expect to see a corresponding change in resource allocation and the policy. To depict this change graphically, we compute the entropy of the agent's

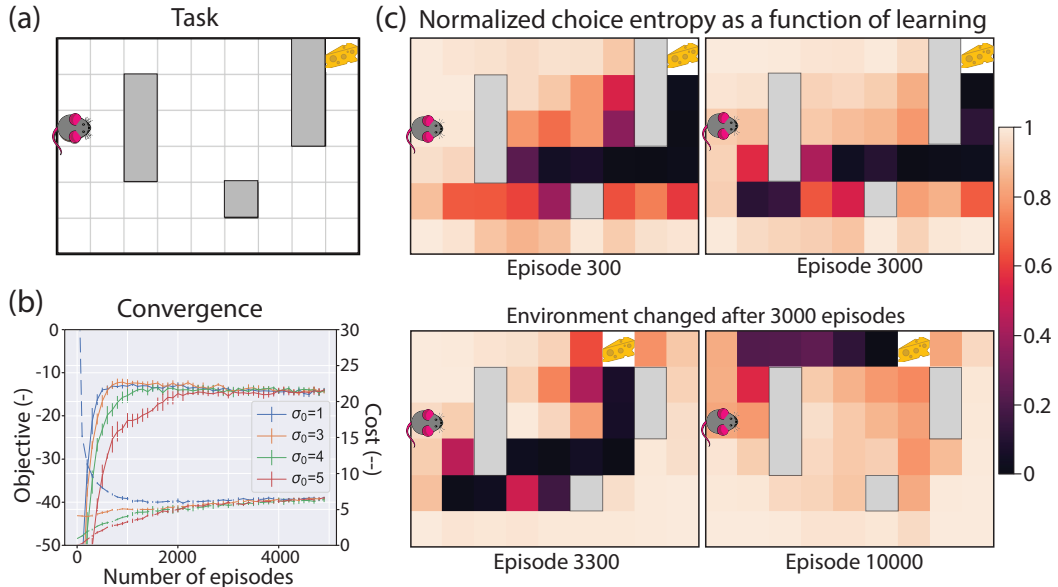

Figure 1: 2D grid-world. (a) Task (adapted from [16]). (b) Rate of convergence of the objective $\mathcal{F}$ (solid lines, Eq. 1) that the agent aims to maximize and the cost (dashed lines) it pays to encode memories with higher precision. Error bars represent SD across mean of 5 optimization runs. (c) Normalized entropy of the agent's policy through training and re-training after change in the environment. Darker colors indicate lower entropy and thus more precise memories.

policy at each state during different stages of training. Namely, we compute the choice probability vector as the normalized histogram of $10^5$ actions from each state and then compute the entropy of those vectors. In Fig. 1c, memories corresponding to states to which DRA allocates more resources are indicated by lower entropy, *i.e.* less randomness, of the policy. Early on, the agent's memories get more precise for states that are close to the reward (top-left panel), whereas an over-trained agent only remembers the shortest path to the reward (top-right panel). Moreover, immediately after the change in the environment, we expect agents to follow the previously optimal path (top-right panel) and turn left at the end, and eventually follow the shorter paths. The bottom panels show that agents learn to reallocate resources to better paths over time but they still retain traces of the older memories.

We also compare DRA against a model that allocates resources equally to all memories ('equal-precision'), but the precision shared across all memories is otherwise subject to the same optimization procedure as DRA. We find that DRA achieves $2\times$ improvement in the objective (Eq. 1) over the equal-precision model. Finally, we construct another baseline model by letting $\lambda \to 0$ in DRA, which reduces it to SARSA, and report that DRA only takes $1.5\times$ the number of episodes to converge as compared to the baseline model while making efficient use of memory resources. Similar findings hold for all the tested hyperparameters in a wide range (see Appendix B.2).

## 3.2 Mountain car

Next, we test DRA on the mountain car problem [39], where an under-powered car needs to reach the flag on top of the hill (Fig. 2a). At each time-step, the car can accelerate *left* or *right* by a fixed amount, or do *nothing*. It always starts at the bottom of the hill $x = -0.5$ with velocity $v = 0$ and must swing left and right, gaining momentum to progressively reach higher. In Figs. 2b-c, we show the mean of the value function for each state, computed as $\bar{v}(s) = \max_a \bar{q}(s, a)$, as well as the entropy of the agent's policy for each state computed as described in Section 3.1. We find that DRA sensibly encodes memories corresponding to states that are close to the states in the optimal trajectory for this task with higher precision.

Further, in Fig. 2d we show the performance of alternative gradients (Eq. 4) that may be used to allocate resources with DRA. Our results are in line with previous work in that the advantage gradient

outperforms other approaches [32]. Finally, we perform the same analyses as in Section 3.1 by comparing DRA against an 'equal-precision' model that allocates resources equally to all memories, where DRA achieves a $1.3\times$ improvement in the objective; and by comparing DRA against a baseline model ($\lambda = 0$), where DRA takes $1.3\times$ the number of episodes to converge while making efficient use of memory resources.

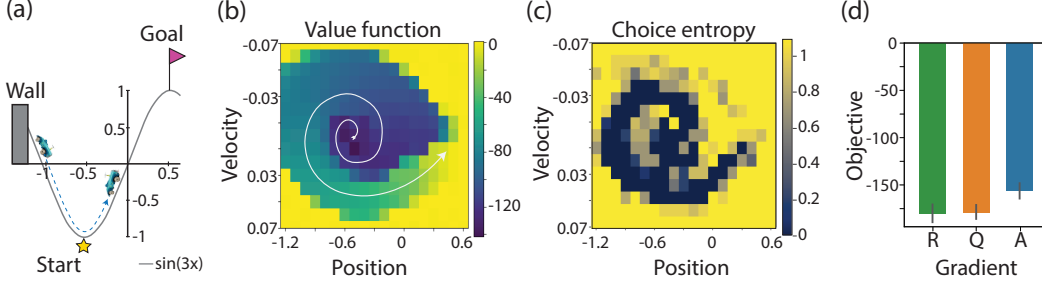

Figure 2: Mountain car. (a) Task. (b) Value function learnt by the agent with a close-to-optimal route indicated by the white arrow. (c) Entropy of the agent's policy indicating precision of corresponding memories. (d) Maximum objective achieved by using the Advantage function (A), the mean q-value (Q), and REINFORCE (R) to compute the stochastic gradient of the first term of $\mathcal{F}$ – the expected reward (Eqs. 1-4). Errorbars represent SD of the mean across 5 optimization runs.

## 4 Results on a model-based planning task

### 4.1 Task details

To study the effect of resource allocation in model-based planning, we consider here the task devised by Huys et al. [40], whose deterministic state transitions and immediate rewards are described in Fig. 3a. Participants never see this underlying structure, but are trained extensively on the transition structure and immediate rewards until they pass a test. Subjects are asked to perform $M \in \{3, 4, 5\}$ moves on each trial, indicated in advance, with the goal of maximizing cumulative rewards. Crucially, the subjects must plan their sequence of moves in a fixed time-period of 9s, during which they cannot act, and get a subsequent 2.5s to execute the entire sequence of actions. With perfect knowledge of the task, the optimal sequence can be found by simply exploring all $2^M$ sequences and selecting the one associated with the highest reward (*e.g.*, from state 5, with $M = 3$, one should pick state 6, 1, & 2). Because of the time pressure, however, subjects are unable to explore all possible moves.

We construct an MDP for this task by expanding the state space by a factor of $M$, allowing the task to be Markovian. In this section, we consider the MDP for $M = 3$ moves, but the results hold true generally (see Appendix B.1). Agents thus have $N_{\text{mem}} = |\mathcal{S}| \times M \times |\mathcal{A}| = 36$ memories of the form $\langle s, a, s', r, \mathcal{N}(\bar{q}_{sa}, \sigma^2_{sa}) \rangle$, with perfect recollection of $s', r$ given $s, a$, but without prior knowledge of the q-values which are initialized randomly around $\bar{q}(s, a) = 0 \ \forall s, a$ with some precision $\sigma(s, a) = \sigma_0 \ \forall s, a$.

In order to plan, agents start in the initial state $s_0$ and choose next states according to their policy until they reach a terminal state, which is when they reset $s = s_0$ and continue planning. Agents keep track of all paths traversed and rewards accumulated for each path until they reach the time limit, which we implement by limiting the number of states that the agent can explore: the *search budget*. When this search budget is exhausted, agents choose the sequence of actions corresponding to the most-rewarding path in their working memory as shown by the schematic in Fig. 3b.

### 4.2 Comparison with an alternative model and black-box optimization

We compare the results obtained via our gradient-based resource allocation to a black-box optimization procedure (CMA-ES, described in Section 2.3). For both methods, we fix the mean of the q-value distribution to the optimal point-estimate $\bar{q}^*$ obtained via q-learning [3], and maximize the objective $\mathcal{F}$ (Eq. 1) with respect to the resources, $\boldsymbol{\sigma}$. Fig. 3c shows that both methods perform comparably when we sample $N_{\text{traj}} \geq 10$ trajectories to estimate the stochastic gradient of $\mathcal{F}$ (see Section 2.3).

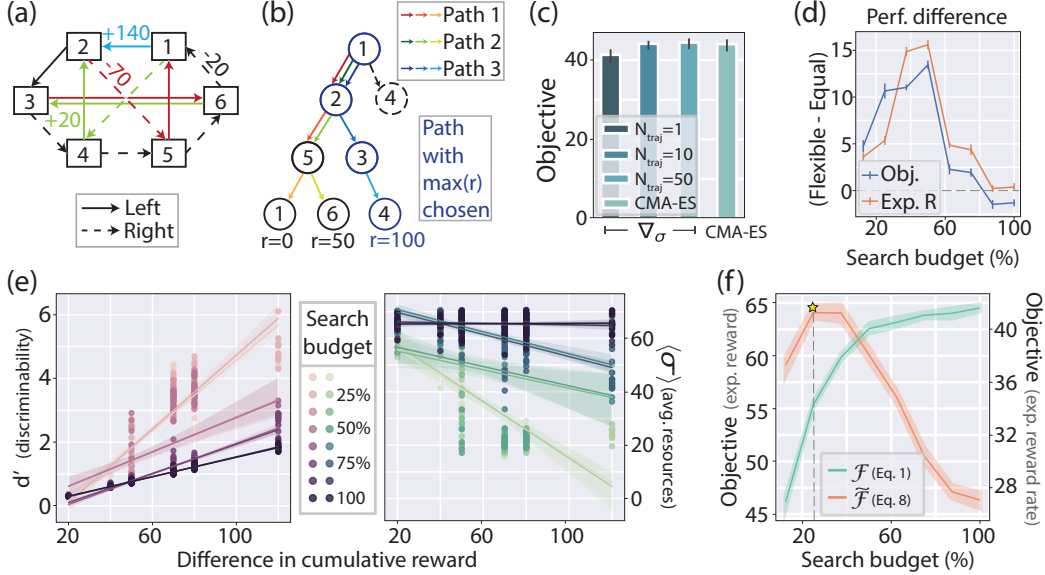

Figure 3: Planning task. (a) Task structure. (b) Paths explored and chosen in an example trial. (c) Comparison of the optimal objective $\mathcal{F}_* = \arg\max_\sigma \mathcal{F}$ found by estimating the stochastic gradient $\nabla_\sigma \mathcal{F}$ with $N_{\text{traj}} \in \{1, 10, 50\}$ sampled trajectories and CMA-ES. (d) Difference in the optimal objective (blue) found by DRA from the optimal objective found by a model that is constrained to have equally precise memories; orange is the same for expected reward. As the search budget increases (planning-time pressure decreases), the advantage diminishes. (e) Linear regression fits for the discriminability of memories (d', left) and average resources allocated to memories (right) as a function of their impact on cumulative reward. As the search budget increases, DRA gives up differential allocation of resources to items in memory as it is no longer advantageous. (f) $\mathcal{F}_*$ (green, left axis) and $\widetilde{\mathcal{F}}_*$ (orange, right axis) as a function of search budget. Errorbars/shaded areas represent SD of mean across 5 optimization runs in (c),(d), and (f), and across memories in (e).

Our agent is limited in its precision of q-values, but only some of them need to be encoded precisely, namely the ones associated with decisions that have a high impact on cumulative rewards, *e.g.* for $M = 2$, $s_0 = 6$, moving to state 1 vs. 3 results in a large difference (120 points) in cumulative reward. To show that it is indeed beneficial to prioritize some memories over others by encoding them with higher precision, we also trained an agent that is constrained to have all its memories equally precise ('equal-precision'), though how precise is subject to the same optimization procedure. Fig. 3d shows the advantage in performance of the agent that allocates resources flexibly (DRA) from the one that is constrained to allocate resources equally. This advantage is more pronounced when the agent is under time pressure (lower search budget) to plan its sequence of moves as expected.

Furthermore, this task reveals that the key factor for resource allocation is not memory precision per se, but the *discriminability* d' (a ratio of difference in means divided by the effective standard deviation) between q-values of a state. In Fig. 3e, we see this effect manifested strongly when agents have a low search budget, and it flattens as the search budget increases to 100%, *i.e.* when agents can explore all possible paths, such that the reward they obtain is unaffected by the precision of q-values. Since d' is correlated with difference in cumulative reward, we also plot the mean $\sigma$ of memories (across actions) to show that this effect indeed stems from resource allocation. Intuitively, DRA allocates more precision to memories when larger differences in rewards are at stake (Fig. 3e).

## 4.3 The speed-accuracy trade-off

So far, we have restricted our analyses to cases where agents are given a fixed search budget to plan their moves, and they optimize their objective $\mathcal{F}$ independently for each fixed budget. As expected, their performance increases monotonically with the time they spend planning and eventually saturates as shown in Fig. 3f (green curve). In most real-world scenarios, however, the search budget is unknown and therefore agents need to decide when to stop planning and execute an action. We can

incorporate this speed-accuracy trade-off [30, 41] in our framework by modifying the objective $\mathcal{F}$ as:

$$\widetilde{\mathcal{F}} := \frac{\mathbb{E}_\pi \left[ \max_{\text{path}=1}^{bM} \left( \sum_{t=1}^{M} r_{t,\text{path}} \right) \big| Q \right]}{\langle a_{\text{dec}} b + t_{\text{non-dec}} \rangle} - \lambda D_{\text{KL}} \left( Q \parallel P \right) \qquad (8)$$

where the numerator of the first term represents the expected reward from planning specific to this task as described in Section 2.3 and Fig. 3b, and the denominator represents the average time the agent spends per trial written as the sum of the average decision-time (proportional to the search budget $b$ with proportionality constant $a_{\text{dec}}$) and non-decision time $t_{\text{non-dec}}$ (*e.g.* due to sensory delay and executing motor output). In many cases such as in this task, agents can follow locally estimated gradients with respect to $b$ as they allocate resources dynamically with DRA to maximize $\widetilde{\mathcal{F}}$ since this turns out to be a convex optimization problem (Fig. 3f, orange curve).

Note that in this analysis, agents draw single Thompson samples from all the memories at each state during planning (see Eq. 2). In principle, they could spend more time at some states than others, in which case, the effective precision of memories would not only be controlled by the number of neurons, but also the time available to plan. However, a detailed analysis of such flexible planning is outside the scope of the current work.

## 5  Discussion

In this article, we propose a framework to model uncertainty in action-values stored in limited memories, show how resource-limited agents can plan and act with such uncertainty, and how they benefit by prioritizing memories differentially. Our work provides a novel, normative approach to dynamically allocate limited memory resources for finding good policies for decision-making. Though we only consider action-values that are independently corrupted by normally distributed noise, our framework can easily be extended to arbitrary choices of noise distributions imposed on values and arbitrary resource costs.

Previous work has considered prioritization of memory access guided solely by the value of backups that lead to a change in the agent's policy [16]. However, such a form of prioritization predicts that all memories are equally relevant when animals are well-trained, which contradicts empirical findings [42, 43]. In order to model and understand animal behavior, it is important to consider irreducible sources of uncertainty in the value function besides learning, due either to resource limitations or stochastic rewards and dynamics [19, 44]. In future work, we would like to combine all these sources of uncertainty in order to make experimentally testable predictions for animal behavior.

Our work partly explains a commonly reported phenomenon in neuroscience: while early sensory and somatosensory brain areas recruit more neurons over the course of training, other brain areas responsible for higher-level cognitive processes such as accessing and storing memories (prefrontal cortex) and evidence integration during decision-making (posterior parietal cortex), either commit more neurons or show higher levels of activity during earlier stages of learning than later stages when animals are well-trained [37, 38, 45]. By showing that resource-constrained agents can accelerate learning by starting with more resources (Fig. 1b), we provide a plausible hypothesis for this observation.

Finally, our framework also makes clear predictions about how action-values should be represented probabilistically in the brain. The details of the predictions will depend on the nature of neural code for probability distributions. Given that action-values are scalar variables, probabilistic population codes – in which each neuron or sub-population is tuned to a specific action-value – provide a biologically plausible neural code, for which there exists strong experimental evidence [46, 47]. For this type of code, the amplitude of the neuronal response encoding a specific action-value should be inversely proportional to the variance associated with this action-value [46]. Such predictions could be tested in animals trained on a foraging task while recording in sensorimotor areas like LIP [24], the superior colliculus [48, 49], or the basal ganglia [25] where neurons are known to encode both actions and expected rewards. One might also imagine that, throughout the course of learning, the number of neurons encoding action-values, or the information-limiting correlations among these neurons [50], are modulated so as to reflect the precision with which these action-values are encoded.

## Broader Impact

We believe that this work has the potential to lead to a net-positive change in the reinforcement learning community and more broadly in society as a whole. Our work enables researchers to represent the uncertainty in memories due to resource constraints and perform well in the face of such constraints by prioritizing the knowledge that really matters. While our work is preliminary, we believe that furthering this line of work may prove to be highly beneficial in reducing the overall carbon footprint of the artificial intelligence (AI) industry, which has recently come under scrutiny for the jarring energy consumption of several common large AI models that produce up to five times as much $CO_2$ than an average American car does in its lifetime [51, 52].

In terms of ethical aspects, our method is neutral per se. The advancement of energy-efficient algorithms may enable autonomous agents to function for long hours in remote areas, the applications for which could be used for both constructive and destructive things alike, *e.g.* they may be deployed for rescue missions [53] or weaponized for military applications [54, 55], but this holds true for any RL agent.

## Acknowledgments and Disclosure of Funding

We thank Pablo Tano, Reidar Riveland, Rex Liu, and David Redish for useful discussions, and Morio Hamada for providing helpful feedback on a previous version of the manuscript.

Nisheet Patel was supported by the Swiss National Foundation (grant number 31003A_165831). Luigi Acerbi was partially supported by the Swiss National Foundation (grant number #31003A_165831) and by the University of Helsinki (Faculty of Science), through grants of the Academy of Finland. Luigi Acerbi also thanks the Academy of Finland Flagship programme: Finnish Center for Artificial Intelligence (FCAI). Alexandre Pouget was supported by the Swiss National Foundation (grant numbers 31003A_165831 and 315230_197296). The funders had no role in study design, data collection and analysis, decision to publish, or preparation of the manuscript.

## Footnotes

[1]Code to run DRA and reproduce our results is available at `https://github.com/nisheetpatel/DynamicResourceAllocator`.

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
