[Supplementary Material]

# Dynamic allocation of limited memory resources in reinforcement learning – Appendix

**Nisheet Patel**[*]
Department of Basic Neurosciences
University of Geneva
nisheet.patel@unige.ch

**Luigi Acerbi**
Department of Computer Science
University of Helsinki
luigi.acerbi@helsinki.fi

**Alexandre Pouget**
Department of Basic Neurosciences
University of Geneva
alexandre.pouget@unige.ch

## A   Computing the gradient to maximize the objective function

### A.1   Gradient of the log policy

In order to compute $\nabla_\sigma \log(\pi(a|s))$, we first note that we can rewrite draws from the memory distribution, $\tilde{q}_{sa} \sim \mathcal{N}(\bar{q}_{sa}, \sigma_{sa}^2)$, as $\tilde{q}_{sa} = \bar{q}_{sa} + \zeta_{sa}\sigma_{sa}$, where $\zeta_{sa} \sim \mathcal{N}(0,1)$ [1]. In this section, we abuse the notation slightly to omit the explicit dependence on the state-action pair $(s,a)$ for clarity, and instead place it in the subscript. With this, we can write our policy $\pi$ as a probability vector for all actions $a$ in a given state $s$:

$$
\begin{aligned}
\pi(a|s) &= \delta\big(a, \arg\max_{a'}(\bar{q}_{sa'} + \zeta_{sa'}\sigma_{sa'})\big) \\
\pi(\cdot|s) &= \lim_{\beta\to\infty} softmax(\bar{\boldsymbol{q}}_s + \boldsymbol{\zeta}_s\boldsymbol{\sigma}_s, \beta) \\
&= \lim_{\beta\to\infty} \frac{1}{\sum_a \exp\beta(\bar{q}_{sa} + \zeta_{sa}\sigma_{sa})}
\begin{bmatrix}
\exp\beta(\bar{q}_{sa_1} + \zeta_{sa_1}\sigma_{sa_1}) \\
\vdots \\
\exp\beta(\bar{q}_{sa_n} + \zeta_{sa_n}\sigma_{sa_n})
\end{bmatrix},
\end{aligned} \tag{A.1}
$$

where in the first line we applied the Thompson sampling rule (that is, pick the action with maximal sampled value), in the second line we rewrote it as the limit of a *softmax* with inverse temperature $\beta \to \infty$, and in the last line we wrote the *softmax* explicitly (as a vector for each entry of $\pi(\cdot|s)$).

Next, we relax the limit $\beta \to \infty$ in Eq. A.1 so as to differentiate the logarithm of the policy $\log \pi$ for $\beta > 0$ with respect to the relevant elements of the resource allocation vector $\sigma(s,a)$ as follows:

$$
\begin{aligned}
\frac{\partial}{\partial\sigma_{sa}} \log\pi(\cdot|s) &= -\frac{\partial}{\partial\sigma_{sa}} \log\Big(\sum_a \exp\beta(\bar{q}_{sa} + \zeta_{sa}\sigma_{sa})\Big) + \frac{\partial}{\partial\sigma_{sa}}
\begin{bmatrix}
\beta(\bar{q}_{sa_1} + \zeta_{sa_1}\sigma_{sa_1}) \\
\vdots \\
\beta(\bar{q}_{sa_n} + \zeta_{sa_n}\sigma_{sa_n})
\end{bmatrix} \\
&= -\frac{\exp\beta(\bar{q}_{sa} + \zeta_{sa}\sigma_{sa})}{\sum_a \exp\beta(\bar{q}_{sa} + \zeta_{sa}\sigma_{sa})}\beta\zeta_{sa} + \beta\zeta_{sa}\delta(a, a_i) \\
&= \beta\zeta_{sa}\big(\delta(a, a_i) - \pi(a|s)\big)
\end{aligned} \tag{A.2}
$$

---

[*]Current address: Département des neurosciences fondamentales, Université de Genève, CMU, 1 rue Michel-Servet, 1206 Genève, Switzerland. Alternative e-mail: nisheet.pat@gmail.com.

where the final step follows from rewriting the *softmax* function as the (soft) policy $\pi(a|s)$ in Eq. A.1, *i.e.* with some $\beta > 0$ but not $\beta \to \infty$.

Thus, the gradient of the logarithm of the policy $\log \pi(a|s)$ with respect to the resource allocation vector $\boldsymbol{\sigma}$ can be written as:

$$\frac{\partial}{\partial \sigma_{s'a'}} \log \pi(a|s) = \begin{cases} \beta \zeta_{sa}(1 - \pi(a|s)) & \text{for } s' = s, a' = a \\ -\beta \zeta_{sa'} \pi(a'|s) & \text{for } s' = s, a' \neq a , \\ 0 & \text{for } s' \neq s \end{cases} \tag{A.3}$$

which is reported as Eq. 5 in the main text.

### A.2 Gradient of the cost

In this section, we show how to compute $\nabla_\sigma D_{\text{KL}}(Q \parallel P)$, where $Q = \mathcal{N}(\bar{\boldsymbol{q}}, \boldsymbol{\sigma}^2 I)$ and $P = \mathcal{N}(\bar{\boldsymbol{q}}, \sigma_{\text{base}}^2 I)$, and $I$ is the identity matrix. Since the covariance matrix is diagonal, we can take the gradient with respect to elements of the resource allocation vector $\boldsymbol{\sigma}$ individually. In other words, we can take the gradient of each memory's marginal normal distribution with its standard deviation:

$$\frac{\partial}{\partial \sigma} D_{\text{KL}}\left(\mathcal{N}(\bar{q}, \sigma^2) \parallel \mathcal{N}(\bar{q}, \sigma_{\text{base}}^2)\right) = \frac{\partial}{\partial \sigma} \mathbb{E}_Q\left[\log\left(\frac{Q}{P}\right)\right]$$

$$= \frac{\partial}{\partial \sigma} \mathbb{E}_Q[\log(Q)] - \frac{\partial}{\partial \sigma} \mathbb{E}_Q[\log(P)]. \tag{A.4}$$

We can expand the first of the two terms as:

$$\frac{\partial}{\partial \sigma} \mathbb{E}_Q[\log(Q)] = \frac{\partial}{\partial \sigma} \mathbb{E}_{x \sim \mathcal{N}(\bar{q}, \sigma^2)}\left[\log\left(\frac{1}{\sqrt{2\pi\sigma^2}} \exp -\frac{1}{2}\left(\frac{x - \bar{q}}{\sigma}\right)^2\right)\right]$$

$$= \frac{\partial}{\partial \sigma} \mathbb{E}_{x \sim \mathcal{N}(\bar{q}, \sigma^2)}\left[-\frac{1}{2}\log(2\pi) - \log(\sigma) - \frac{1}{2}\left(\frac{x - \bar{q}}{\sigma}\right)^2\right]$$

$$= -\frac{1}{2}\underbrace{\frac{\partial}{\partial \sigma} \log(2\pi)}_{0} - \frac{\partial}{\partial \sigma}\log(\sigma) - \frac{1}{2}\frac{\partial}{\partial \sigma} \mathbb{E}_{x \sim \mathcal{N}(\bar{q}, \sigma^2)}\left[\left(\frac{x - \bar{q}}{\sigma}\right)^2\right]$$

$$= -\frac{1}{\sigma} - \frac{1}{2}\frac{\partial}{\partial \sigma} \mathbb{E}_{z \sim \mathcal{N}(0,1)}\left[z^2\right]$$

$$= -\frac{1}{\sigma} \tag{A.5}$$

where we use the variable transformation $z = (x - \bar{q})/\sigma$ in the penultimate step. We can follow a similar approach for the second term to get:

$$\frac{\partial}{\partial \sigma} \mathbb{E}_Q[\log(P)] = \frac{\partial}{\partial \sigma} \mathbb{E}_{x \sim \mathcal{N}(\bar{q}, \sigma^2)}\left[\log\left(\frac{1}{\sqrt{2\pi\sigma_{\text{base}}^2}} \exp -\frac{1}{2}\left(\frac{x - \bar{q}}{\sigma_{\text{base}}}\right)^2\right)\right]$$

$$= \frac{\partial}{\partial \sigma} \mathbb{E}_{x \sim \mathcal{N}(\bar{q}, \sigma^2)}\left[-\frac{1}{2}\log\left(2\pi\sigma_{\text{base}}^2\right) - \frac{1}{2}\left(\frac{x - \bar{q}}{\sigma}\right)^2 \frac{\sigma^2}{\sigma_{\text{base}}^2}\right]$$

$$= -\frac{1}{2}\underbrace{\frac{\partial}{\partial \sigma} \log\left(2\pi\sigma_{\text{base}}^2\right)}_{0} - \frac{1}{2}\frac{\partial}{\partial \sigma} \mathbb{E}_{x \sim \mathcal{N}(\bar{q}, \sigma^2)}\left[\left(\frac{x - \bar{q}}{\sigma}\right)^2\right] \frac{\sigma^2}{\sigma_{\text{base}}^2}$$

$$= -\frac{1}{2}\frac{\partial}{\partial \sigma} \mathbb{E}_{z \sim \mathcal{N}(0,1)}\left[z^2\right] \frac{\sigma^2}{\sigma_{\text{base}}^2}$$

$$= -\frac{1}{2}\frac{\partial}{\partial \sigma}\frac{\sigma^2}{\sigma_{\text{base}}^2}$$

$$= -\frac{\sigma}{\sigma_{\text{base}}^2}. \tag{A.6}$$

Combining Eqs. A.4, A.5, and A.6, we can write our analytically obtained gradient of the cost term with respect to individual elements of the resource allocation vector $\sigma(s, a)$ as:

$$\frac{\partial}{\partial \sigma_{sa}} D_{\text{KL}}\Big(\mathcal{N}\big(\bar{\boldsymbol{q}}, \boldsymbol{\sigma}^2 I\big) \parallel \mathcal{N}(\bar{\boldsymbol{q}}, \sigma_{\text{base}}^2 I)\Big) = \frac{\sigma_{sa}}{\sigma_{\text{base}}^2} - \frac{1}{\sigma_{sa}}. \tag{A.7}$$

### A.3 Justification for our choice of the gradient of expected reward

A potential concern regarding our method of allocating resources may be our choice of the advantage function to compute the gradient of the expected rewards (Eqs. 3-4 in the main text). Crucially, the advantage gradient uses the means of the q-value distributions of the relevant memories. However, our main assumption in the paper is that agents do *not* have direct access to the mean of the q-value distribution. According to our assumption, the agent could only estimate the mean by averaging over a large number of samples from the distribution, a process which could take a considerable amount of time (because sequential samples from memory would be highly correlated [2]).

This concern is resolved by considering that in DRA the resource allocation vector is not updated during the trial, but rather only *offline*, *i.e.* before or after the trial, or potentially during sleep. This way, during the task, the agent draws single (Thompson) samples in order to act and does not waste extra time in order to consolidate and reallocate resources across its memories.

While 'offline sampling' resolves the issue of how agents can access the mean of the distribution to compute policy updates, and it is the approach followed in this work, it represents a binary solution (*i.e.*, either the agent takes one Thompson sample online, or a very large number of them offline). We could generalize this approach by allowing an agent to take *multiple* samples from its q-value distribution to get a better estimate of the expected return while performing the task. Taking additional samples would cost them time, which they could potentially use to act in the environment and collect rewards. If the opportunity cost is higher than the potential increase in rewards obtained by taking more samples, they may not want to waste time sampling but instead make their memories (q-value distributions) precise enough that fewer samples suffice to maximize reward given their storage capacity. This is another example of the speed-accuracy trade-off we considered in Section 4.3 in the main text, and which we leave to explore for future work.

## B Task parameters and additional results

### B.1 Additional results for the planning task

In the main article, we showed results for the planning task we adapted from Huys et al. [3] where subjects had to plan sequences of $M = 3$ moves. More generally, we ran DRA for $M \in \{3, 4, 5\}$, showing that the algorithm allocates resources differentially depending on $M$ (Fig. B.1).

Figure B.1: Linear regression fits for the discriminability of memories as a function of their impact on cumulative reward for the planning task with number of moves $M \in \{3, 4, 5\}$.

Table B.1: Parameters used for each task

| Parameter | Task | | |
| --- | --- | --- | --- |
| | Grid-world | Mountain Car | Planning task |
| $\alpha_1$ | 0.1 | 0.1 | 0.1 |
| $\alpha_2$ | 0.1 | 0.1 | 0.1 |
| $\beta$ | 10 | 10 | 10 |
| $\gamma$ | 1 | 1 | 1 |
| $\lambda$ | 0.2 | 0.1 | 1 |
| $\sigma_{\text{base}}$ | 5 | 5 | 100 |
| $\sigma_0$ | 3 | 3 | 50 |
| $N_{\text{traj}}$ | 10 | 10 | 10 |
| $N_{\text{restarts}}$ | 5 | 5 | 5 |

## B.2 Task parameters

In this section, we report (Table B.1) and briefly describe the (hyper-)parameters chosen for each task. For the present study, we fixed the learning rates for the means and standard deviations of the memory distribution, $\alpha_1$ and $\alpha_2$ respectively, to reasonably low values. We set the inverse temperature parameter $\beta$ to a reasonably high value (for the *softmax* approximation to the 'hard' max to hold, as per Section A.1), but not too high to restrict the influence of individual updates on the resource allocation. As mentioned in the main text, we exclude discounting for all the tasks and thus set $\gamma = 1$. Perhaps the most important choice is the parameter $\lambda$ that introduces a trade-off between the expected reward and the cost of being precise. We chose $\lambda$ that we best captured the difficulties faced by memory-limited agents, but a range of nearby values yields qualitatively similar results, *e.g.* in the mountain car task, $\lambda \in [0, 0.4]$ allows agents to perform the task well with enough training. The other equally important parameter would perhaps be $\sigma_{\text{base}}$, which would represent the resources for some base distribution of q-values in memory before training. $\sigma_{\text{base}}$ controls how discriminable different actions would be from a given state, and we chose it appropriately given the reward structure of each task. As mentioned in the main text, starting with a higher resource budget than the base distribution, *i.e.* with $\sigma_0 < \sigma_{\text{base}}$, either by means of paying more attention or allocating more neurons, allows agents to accelerate learning. We sampled $N_{\text{traj}} = 10$ trajectories to update the resource allocation vector at the end of each trial with adequate precision. As shown in Fig. 3c in the main text, sampling more trajectories does not yield better performance, but less leads to variability in the stochastic estimate of the gradient and thus hurts performance. Finally, we performed $N_{\text{restarts}} = 5$ optimization runs for each task to report an estimate of variability across runs (*i.e.*, error bars), as mentioned in the main text. In addition to the above parameters used to display the results, we systematically varied the values of $\lambda$ and $\sigma_{\text{base}}$ in all tasks and report that the qualitative results hold for a large range of values of these parameters with $\sigma_{\text{base}}$ having a slightly stronger effect than $\lambda$.