[Reviews · NeurIPS 2020]

Review 1

Summary and Contributions: This work proposes an algorithm for dynamically modulating the precision of "memories" in a reinforcement learning agent. The algorithm, DRA, uses stochastic gradient descent to optimize an objective that maximizes expected reward given the memory, and minimizes the difference between the memories and their prior. The algorithm is tested on a few probe tasks, and the authors note some interesting empirical observations; namely, that this algorithm learns to more precisely encode "important" memories. Finally, the authors suggest that this approach offers a normative solution to the problem of learning how to dynamically allocate resources, although the actual process of resource allocation was not addressed here. Altogether, the experiments are conducted well, the writing and ideas are crisp, and this work is a nice addition to this space of ideas.

Strengths: This work is very well motivated. It has some obvious biological inspiration, tackling the question of resource allocation from the perspective of thinking how brains might accomplish this via learning. The writing is crisp, and the paper was a pleasure to read. I was rarely looking for more details, and the ideas were easy to understand even after a single read. While the experiments are stripped down, they more-or-less accomplish what they are designed to do. This is not a paper searching for state of the art results, and it should not be treated as such; rather, it is an exposition of a particular idea, and it did well to explore it. While I do not think the work is significant or groundbreaking in any way, I don't think this detracts from its value.

Weaknesses: (This section is being combined with "comments for improvement" section below) A bit of a nitpick regarding the language use around “more” or “less” resources. The authors write about an agent using “more resources”, which corresponds to “lower entropy” for the actions in a particular state. I think, though, that technically (and literally, for this agent) the amount of resources used for each memory is exactly the same; it’s literally a number to represent the mean and standard deviation. From what I can tell, the authors are arguing that memories with lower standard deviations would *require* more resources to represent in certain implementations (such as in brains). So it’s not actually the case that more resources are used for low-sigma memories in the agent, but that more resources might be used in other agents. It might be more accurate to talk about memory *precision* instead when reporting results (the authors do this, but they also go back and forth between talking about “resources”). So another way of interpreting the algorithm here has to do with how precise we allow memories to be in the agent, and then we can make it explicit that more precise memories will require more resources to encode in certain implementations. The authors do indeed speak to these ideas throughout the paper, but I think they can be made a bit more explicit, and when it comes to actually implementing the algorithm in the agent as done here, it might be more accurate to talk about memory precision rather than memory resource-amount, and/or necessarily equating the two. Certainly in computer implementations of this algorithm there isn’t a strict correspondence between precision and resources. I wonder if the authors have any data that show the values of sigma prior to and after learning, especially in the case where they vary the initial sigma values and show different convergence speeds? Is it the case that the final sigma values are all identical across the different hyperparameter choices for initial sigmas, or do the different initial choices converge to different final values? The normalized entropy is shown in one figure, but this cannot be directly related to the initial sigma values picked. I ask for a few reasons: the initial sigma values are more-or-less arbitrary and effectively represent a hyper-parameter. Choosing sigma=1 is no more “principled” than choosing sigma=5, and if the sigmas converge to, say, 0.001 by the end of learning then the differences in learning speeds shown might better reflect our choice of hyperparameters than say anything meaningful about the effect of resource allocation/memory precision. Another possible case is if the sigmas converge to a very low value, indicating that the algorithm ultimately “tries to” implement an infinitely precise q-lookup table, in which case this algorithm would be “getting in the way of” that. If the opposite were true (that is, high-sigma memories learn faster than low-sigma memories) then it would be less likely that this would be true, since there would be an advantage to having less-precise memories early on. I think it is also important for the authors to include standard baseline implementations so we can gauge the effectiveness of this algorithm compared to known methods. I don't request this because I'd like to see their algorithm "beat" baselines, but rather, because it is an important empirical fact whether DRA is, say, 10x slower, 1000x slower, 100x faster, etc. This can have important implications for whether we expect the algorithm to scale. What happens if the memories approach infinite precision? Does this implement a standard q-table and/or revert DRA to some form of q-learning? Can this baseline be included?

Correctness: As far as I can tell everything is technically sound, thought I may not be the best judge for rigourous checking of the proofs and derivations.

Clarity: Yes, it is very well written.

Relation to Prior Work: Related work is clearly discussed. I think that this topic is actually quite broad, though, so I can imagine that much related work was not mentioned. For example, there has been some work on dynamic allocation of memories (though not explicitly for the purposes of resource management, it can be interpreted as such). I don't think lack of citation here is necessarily an issue since I'm convinced that the authors sufficiently corroborate their claims and give a broad enough background, but it might be true that it's impossible to truly cover the space of past work.

Reproducibility: Yes

Additional Feedback: Please see the other sections for major comments. Altogether I think this is a solid piece of work that can be further improved by refining the language and by including more baseline results.


Review 2

Summary and Contributions: The paper introduces an RL framework which allows the agent to dynamically allocate importance (precision) to memories. Memories (q-values) are represented as distributions from which the agent can sample rather than point estimates. The contributions of the paper are as follows: - Introducing a new biologically-inspired memory mechanism to RL. - New cost functions which account for limited memory resources. - The key contribution with respect to the existing work is providing the RL agent with control (through a resource allocation vector) over the precision of its memories, which is grounded in research on memory allocation in biological brains. - Experiments on the task of finding the shortest route in a 2D grid-world which show that starting with more resources leads to faster learning. - Evaluation on a model-based planning task. A comparison with a model of equally precise memories, which shows that it is beneficial for an agent to prioritize some memories over others. The authors also show that their proposed framework (DRA) allocates more precision to memories that have a high impact on cumulative rewards.

Strengths: The work is bridging parallel lines of research on allocation of limited memory resources in neuroscience and reinforcement learning, and it contributes to the discussion on bringing RL agents closer to the biological ones by introducing the limited memory constraint. The proposed memory allocation algorithm is argued to exhibit similar behaviour to frontal cortical areas in a biological brain. Additionally, the work is focused on model-based RL which is of high interest to the community. In terms of novelty, relevance and potential impact, it is an appropriate paper for NeurIPS.

Weaknesses: The results from the mountain car experiments have a limited analysis and interpretation.

Correctness: To the best of my knowledge, definitions of the environment, assumptions, objectives and policies are sound. Experimental methodology is correct and it accounts for high variance in RL algorithms.

Clarity: The paper is clear and it reads very well. The limitations of the proposed method (for instance, considering only normally distributed and independently encoded action-values) are discussed. At every step, the authors draw parallels between their work and neuroscience research and/or provide explanations for the choices they make (for instance, using REINFORCE in place of advantage function in planning, lines 119-126). The environments and tasks studied in the experimental sections are described from scratch which makes the contributions easier to appreciate.

Relation to Prior Work: There is a thorough comparison with prior work (lines 46-60). I would suggest a separate section on related work rather than a paragraph in the submission, and adding the discussion on previous ways of prioritizing memory access (Mattar and Daw 2018) to the new Related work section.

Reproducibility: No

Additional Feedback: How to generalize this approach to arbitrary q-value distributions without the assumption of memory independence? How would it work on continuous tasks?


Review 3

Summary and Contributions: Most RL algorithms have assumed that agents can access and update the state values or action values in infinite precision, which is biologically infeasible. This study proposes an algorithm that allows agents to control the precision of memories so that the agents could maximize the cumulative rewards while minimizing the computational costs for memory allocations. This provides a more efficient and biologically feasible learning mechanism which is important for model-based RL and future planning.

Strengths: To incorporate inherent uncertainty in RL, this study makes the agent able to represent task structure with the distribution of action values which is different from the discrete representation by allowing an agent to represent the level of precisions according to its importance as well as the mean values. To make this computable, this study decoupled the time for value updates and the time for mapping the value distribution. In the aspect of using the idea of prioritized replays in RL, this study enables agents to allocate resources specifically for the current task which is distinguished from the previous studies of efficient resource allocation. This algorithm is tested not only for spatial navigation but the abstract task required planning which suggests that this algorithm can be generalized for broader RL problems.

Weaknesses: It would be nice if authors can provide more details on how N trajectories were generated. 1) Are N trajectories similar concepts of replay in the Dyna model? 2) Does each of N trajectories always start from the current state (S_t)? If so, is it true that the forward trajectories (S_t -> the goal ) are more likely to be replayed (it is possible to have a backward trajectory but it is not useful when there is directionality such as 2D grid world example)? Previous studies suggest that the backward replay allows agents to prioritize credit assignment. Is this algorithm also used to prioritize resource allocation for efficient credit assignment by extending from future planning? 3) What is the condition of termination of each trajectory? When N=10, do the agents continuously find the trajectories until having 10 successful trajectories that make them reach the goal? If so, the agent sample N trajectories at every state (S 1, ... S goal-1)?

Correctness: It is confusing whether Fig.1b and Fig.1C were made from the same or different simulations. If the reward location and environment were changed in episode 3000 in Fig.1b, there is no big difference in adaptation speed according to 𝜎. If they are coming from different simulations, please specify explicitly. Can the authors also compare the changes in the objective between the flexible allocation and the equal allocation for 2D grid world simulation (like Fig.3b)? It is important to show that an agent has a better memory of a state not because of the frequent visits but as a result of efficient resource allocation.

Clarity: Generally well-written. Only the parts may need more explanation are remarked above.

Relation to Prior Work: Yes. After introducing the previous studies in two tracks, 1) how to archive optimal resource allocation, and 2) how to plan optimal route for the future actions in a task space, they discussed what they made differnt from these previous works.

Reproducibility: Yes

Additional Feedback: UPDATE The divergent scores among reviewers seem to come from differences in interests. I agree that this may not provide a state-of-art algorithm to improve an AI agent navigation strategy but I believe that an exploration of biologically plausible RL is potentially important for future studies especially for generalization and flexible decisions. At least on my understanding, though 'the agent can observe and store states and actions perfectly', it is still different from the agent who has a perfect transition structure of the world (at least in the grid example). It is more likely to know one state and the following state in the given action but not the structural map. It may be different in the planning-task in which arriving at a specific state is not the goal (unlike the grid world) but finding a path maximizing the cumulative rewards. Therefore, in both cases, the agent makes a decision in the current trial based on previous experiences but cannot make a zero-shot inference and vector-based navigation to the goal. Updating all the Q values from one trial of experience is computationally expensive for the brain (though I agree that it is can be very cheap for the AI agent). To maximize the efficiency in a limited resource, the brain may need the rule to assign a priority of which Q value should be processed first. Regarding the issue raised by other reviewers that some readers may hard to find the interests in the topic, it could have been great if the authors can show whether this algorithm explains the actual behavior of animals or humans better than alternatives such as a non-prioritizing algorithm. To avoid misunderstanding, the authors should explain the level of the task structure representation of the agent clearly. --- (1) Fig.3f - Which side of y axis range was used to explain the orange and green graph?; (2) The line 201 in page 6 - N has been already used to indicate the number of trajectories to sample.; (3) Eq.8 - Were 'a_dec' and 't_non_dec' were computed by the regression agaisnt the real reaction times? If so, what were the values?


Review 4

Summary and Contributions: Both biological and artificial intelligences have limited computational and memory resources. This paper studies a specific case, where an agent learns an action-value function for every action and state (a tabular setup) but represents the value as a normal distribution and pays a cost proportional to the KL divergence between the distribution and a base distribution with the same mean (this means that there is only a cost for representing the q-value with lower variance (higher accuracy) but there is no cost for representing different means). They derive a policy gradient for this objective and test their algorithm in on 3 tabular tasks (a grid world, mountain car, and a planning task previously used to study human planning under time constraints). They observe that initializing the distributions with lower variance (effectively allocating more initial representational capacity) results in more rapid learning but all initializations converge to similar performance (this is not that surprising since the objective function is the same in all cases, just the initial starting conditions differ). They argue this may explain while more neurons seem to be involved in initial learning in animal experiments than are required once the task is learned.

Strengths: The paper is mostly well-written. There is interest in the community to understand both the connection between biological learning and RL, especially with limited capacity. Given the assumptions, the learning approach appears sound. This particular approach to dealing with limited representational resources is novel as far as I know.

Weaknesses: The intended audience for this work is a bit unclear. If it is intended as a model to understand biological brains better the connections are quite limited and its not clear that it provides much novel understanding. Conversely, since all experiments are on small, tabular environments for which optimal solutions easily fit in even a very limited computer memory, it does not seem of interest to efforts to improve artificial intelligence. The choice of how to limit algorithm capacity seems arbitrary and not well justified (for example, Huys et al. shows humans seem to truncate search trees to avoid negative rewards and build ``options'' and more generally it seems likely that other approaches to limited representational capacity would agglomerate states, so the choice of a perfect tabular representation where each state action pair is captured with perfect fidelity but with limited accuracy to represent the value function seems arbitrary). Indeed the wide use of fixed size parametric function approximators like neural networks for RL is a way of limiting memory and computational resources while trading off perfect value function estimation. The connection is even stronger when considering RL as inference approaches such as SAC [1] with a KL regularizer that penalizes deviation from a prior policy for this network. The empirical evaluation is all on very simple tabular tasks, which seems like the least interesting regime for understanding trade-offs in memory resources. The value of $\lambda$ and $\sigma_{base}$ are not varied within a task, so the results are all informing different initial representational capacity, but not studying trade-offs in the asymptotic performance with differing representational capacity. [1] Soft Actor-Critic: Off-Policy Maximum Entropy Deep Reinforcement Learning with a Stochastic Actor. https://arxiv.org/abs/1801.01290

Correctness: I did not check the full derivation of the policy gradient carefully, but it appears correct, and certainly it seems plausible that there are algorithms to optimize eq (1). The empirical work and theory appear reasonable given the assumptions (the weaknesses of which I addressed above).

Clarity: Overall, the paper is well-written. Figure 3 is very complex figure, it would be helpful to expand the caption to explain the figure and conclusions in more detail.

Relation to Prior Work: This work engages well with prior work in neuroscience or studying memory tradeoffs in particular. But it does not discuss other approaches to optimizing RL with limited capacity such as [1] or approaches with cluster states [e.g. 2] [1] Soft Actor-Critic: Off-Policy Maximum Entropy Deep Reinforcement Learning with a Stochastic Actor. https://arxiv.org/abs/1801.01290 [2] The Laplacian in RL: Learning Representations with Efficient Approximations https://arxiv.org/pdf/1810.04586.pdf

Reproducibility: Yes

Additional Feedback: I am updating this to to note I've read the author's response to reviewers. They helpfully clarified a number of minor issues. The two main (related) weaknesses I raised were: - The assumptions seemed somewhat unjustified. There is not a particular connection to neuroscience that representational capacity is likely to be dealt with by limiting q-value precision but maintaining full fidelity in states and actions. - For this reason the interest of this work to the community seems limited (in that it sets out some assumptions, solve the problem those assumptions lead to, but does not provide a compelling reason why those set of assumptions are of particular interest). The response "In our work, we assume that agents have limited capacity to store (q-)values in their memory, but that they can observe and store states and actions perfectly. This choice is deliberate. It allowed us to develop a normative solution to the problem of allocating resources to items in memory. As far as we know, we are the first ones to provide such a solution." I agree that (as far as I know) this work is the first to set up these assumptions and provide a solution, but my concern is the lack of justification for why these assumptions setting up this situation should be of particular interest. I will agree with the author's rebuttal that other reviewers disagreed with this assessment, but nothing in the rebuttal or other reviews seems to provide a justification so I've left my score unchanged.

[Author Response · NeurIPS 2020]

We thank reviewers R1, R2, R3, and R4 for their constructive and helpful feedback.

**Scope and significance (R4).**   Our work derives the normative solution to the problem of how to dynamically allocate
noisy and limited memory resources for reinforcement learning. This work could have implications for machine learning
in the long run but our intended audience is currently neuroscientists, since memory access is inherently noisy in neural
circuits. We aim to use this work to generate testable predictions as to what should be observed in neural circuits
when learning complex tasks that require memory access. One possibility is that more neurons are devoted to specific
state-action pairs (in parietal cortex or in the basal ganglia, where q-values are putatively encoded) or memory of the
q-values might be sampled for a longer duration when higher precision is warranted, thus modulating the speed-accuracy
trade-off characteristic of decision making with sequential sampling. We aim to explore these ideas in future work.

**Disambiguating DRA from approaches in RL (R4).**   R4 is correct in pointing out that other groups have proposed
alternative approaches to deal with memory limitations in RL, such as using regularization (SAC [A1]), or using neural
networks for representing policy and value functions, and even compressing state representations with graph Laplacian
[A2]. Our work is meant to complement these previous studies. SAC, for instance, directly penalizes the policy entropy
while maximizing reward to encourage exploration. In DRA, we penalize precise representations of (q-)values instead.
The use of Laplacian in RL [A2], on the other hand, hints at yet another problem involving efficient use of memory –
compact representation of states (*e.g.*, *chunking*) – which is something we very much look forward to addressing in
future work. We modified the *Related work* section to discuss and point out how these proposals complement our work.

**Justifying our assumptions (R4).**   In our work, we assume that agents have limited capacity to store (q-)values in
their memory, but that they can observe and store states and actions perfectly. This choice is deliberate. It allowed us to
develop a normative solution to the problem of allocating resources to items in memory. As far as we know, we are the
first ones to provide such a solution. We agree with R4 that it would be important in the future to combine our approach
with alternatives that focus instead on regularizing the policy or compressing states, including heuristic approaches such
as truncation of planning trees, a strategy suggested Huys et al. [A3]. However, we hope that R4 will agree with us and
other reviewers that the existence of other approaches does not take anything away from our contribution: We propose a
normative solution to an important problem that "*is of high interest to the community (R2)*", and which will eventually
make concrete experimental predictions. "*This is not a paper searching for state of the art results, and it should not be*
*treated as such; rather, it is an exposition of a particular idea, and it did well to explore it (R1)*".

**Convergence and baseline (R1).**   Following a suggestion from R1, we found with new analyses that the asymptotic
values of memory precision, $\sigma_*$, are largely independent of the choices of initial value $\sigma_0$. We also found that
convergence speed does not depend on the difference between the initial and optimal asymptotic values. Also, we can
confirm that letting $\lambda \to 0$ reduces DRA to SARSA. As suggested by R1, we pick $\lambda = 0$ as the baseline to compare
convergence speed and found that DRA is $1.5\times$ and $1.4\times$ slower in the grid-world and mountain car tasks respectively.

**Comparison with equal resource allocation (R1, R3).**   We tested DRA against a model that allocates equal resources
to all memories for the grid-world and mountain car tasks (ref. Fig. 3d), and report a $2\times$ and $1.3\times$ improvement in the
objective (Eq.1) respectively with flexible resource allocation (DRA). We added these results to the revised manuscript.

**Sampling trajectories (R3).**   Conceptually, the sampling of memories is similar to Dyna, with the difference being
that instead of randomly sampling individual memories, DRA samples entire trajectories *on-policy* (though they could
also be drawn from stored episodic memories). In DRA, replays are entirely forward. In our current implementations,
we sample trajectories at the end of each trial from the starting state for that trial to termination. In principle, we could
also do so multiple times during the trial, *e.g.* at each state, and not necessarily until termination.

**Generalizing DRA (R2, R4).**   We aim to address non-independent memories and continuous state spaces in future by
considering a Gaussian process prior over the q-values and using GPSARSA [A4] instead of SARSA to update the mean
q-values, and extend to non-tabular settings by incorporating compact state representations, *e.g.*, [A2].

**Response to remaining comments.**   **R1:** We have now clarified the meaning of "more" or "less" resources in the
text, but we insist that our arguments apply to all systems that are restricted to sample from value distributions but
cannot access its mean and precision directly. We depicted the normalized entropy in Fig. 1 simply to ease visualization,
but performed appropriate checks as mentioned earlier. **R2:** The mountain car task was included to demonstrate general
applicability of DRA to arbitrary problems. We now dedicate a section in the main text to *Related work* and discuss [A1,
A2, A5]. **R3:** Figs. 1b & 1c come from different simulations. We have rewritten the caption for Fig. 3 (also suggested
by **R4**) clarifying the y-axis confusion in 3f. We also mention that $a_{\text{dec}}$ & $t_{\text{non-dec}}$ come from simulations. **R4:** In all
tasks, we systematically varied the values of $\lambda$ & $\sigma_{\text{base}}$ and report that the qualitative results hold for arbitrary values of
these parameters with $\sigma_{\text{base}}$ having a slightly stronger effect than $\lambda$.

**Code release**   We intend to release the code on GitHub as soon as the submissions are no longer anonymous.

**References.**   [A1] Haarnoja et al. *ArXiV* 2018. [A2] Wu et al., *ArXiV* 2018. [A3] Huys et al. *PNAS* 2015. [A4] Engel
et al. *ICML* 2005. [A5] Mattar & Daw. *Nat. Neuro.* 2019.


[Meta-Review · NeurIPS 2020]

This paper nicely bridges between neuroscience and RL, and considers the important topic of limited memory resources in RL agents. The topic is well-suited for NeurIPS (R2) as it has broader applicability toward e.g. model-based RL and planning, although this is not extensively discussed or shown in the paper itself. All reviewers agreed that it is well-motivated and written (R1, R2, R3, R4), although R3 did ask for a bit more explanation on some methodological details. It is also appropriately situated with respect to related work (R1, R2, R3) although R2 suggests a separate related works section, and R4 wanted to see more discussion of work outside of neuroscience, focused on optimizing RL with limited capacity. R1 pointed out that perhaps there’s a bit of confusion between memory precision and use of memory resources, as the former is more accurate for agents, the latter perhaps for real brains - ie more precise representations require more resources to encode in the brain, but this seems to be a minor point. R1 also asked to include standard baseline implementations to test for issues such how their model scales compared to other methods. R4 was the least positive, expressing that the contribution to AI is unclear, that the tasks are too easy and wouldn’t be expected to challenge memory resources. Also the connection to neuroscience is a bit tenuous as the implementation doesn’t seem particularly biologically plausible. In the rebuttal, authors argue that this approach will allow them to generate testable predictions regarding neural representations during learning, some of which are already included in the discussion. I find this adequate, but these predictions should maybe be foregrounded more so as to make clearer the neuroscientific contribution. I’m overall quite impressed with how responsive the authors were in their response, including almost all of the requested analyses. I think the final paper, with all of these changes incorporated, is likely to be much stronger, and so I recommend accept.